**Data Availability Statement:** All relevant data are within the manuscript.

**Funding:** This study has been sponsored by Addis Ababa University. The funders had no role in study

# Data quality assessment and associated factors in the health management information system among health centers of Southern Ethiopia

**Mastewal Solomon[1], Mesfin Addise[2], Berhan Tassew[2], Bahailu Balcha****[3]\*, Amene Abebe[3]**

**1** Hadiya Zone Health Department, Shone Town Administration Health Office, Addis Ababa, Ethiopia,
**2** College of Health sciences and Medicine, School of public Health, Addis Ababa University, Addis Ababa, Ethiopia, **3** College of Health sciences and Medicine, School of Public Health, Wolaita Sodo University, Wolaita Sodo, Ethiopia

\* behailubalcha2@gmail.com

## Abstract

### Background

A well designed Health management information system is necessary for improving health service effectiveness and efficiency. It also helps to produce quality information and conduct evidence based monitoring, adjusting policy implementation and resource use. However, evidences show that data quality is poor and is not utilized for program decisions in Ethiopia especially at lower levels of the health care and it remains as a major challenge.

### Method

Facility based cross sectional study design was employed. A total of 18 health centers and 302 health professionals were selected by simple random sampling using lottery method from each selected health center. Data was collected by health professionals who were experienced and had training on HMIS tasks after the tools were pretested. Data quality was assessed using accuracy, completeness and timeliness dimensions. Seven indicators from national priority area were selected to assess data accuracy and monthly reports were used to assess completeness and timeliness. Statistical software SPSS version 20 for descriptive statistics and binary logistic regression was used for quantitative data analysis to identify candidate variable.

### Result

A total of 291 respondents were participated in the study with response rate of 96%. Overall average data quality was 82.5%. Accuracy, completeness and timeliness dimensions were 76%, 83.3 and 88.4 respectively which was lower than the national target. About 52.2% respondents were trained on HMIS, 62.5% had supervisory visits as per standard and only 55.3% got written feedback. Only 11% of facilities assigned health information technicians.

design, data collection and analysis, decision to publish, or preparation of the manuscript.

**Competing interests:** The authors declare that they have no competing interests.

Level of confidence [AOR = 1.75, 95% CI (0.99, 3.11)], filling registration or tally completely [AOR = 3.4, **95% CI (1.3, 8.7)**], data quality check, supervision AOR = 1.7 95% CI (0.92, 2.63) and training [AOR = 1.89 95% CI (1.03, 3.45)] were significantly associated with data quality.

## Conclusion

This study found that the overall data quality was lower than the national target. Over reporting of all indicators were observed in all facilities. It needs major improvement on supervision quality, training status to increase confidence of individuals to do HMIS activities.

## Introduction

Health management information system (HMIS) is one of the six building blocks of health system that integrate data collection, processing, reporting, and use of the information. Globally, the restructuring of health information systems has been an important trend since its declaration in Alma-Ata conference of primary health care as an essential health care strategy in 1978 [1–3]. Developing countries also launched reforms to improve and expand health information systems as a component of health system reform [4]. The HMIS is a major source of information for monitoring and adjusting policy implementation and resource use in Ethiopia [5, 6]. Health Sector Transformation Plan (HSTP) of Ethiopia considers information revolution as one of the four transformation agendas which involves advancement on the methods starting from data collection to the use of information for decision [5, 7].

Data that are accurate, complete and delivered on time to users is an important aspect in healthcare planning, management and decision making but quality of data is frequently assessed as a component of the effectiveness or performance of the HIS; however data quality assessment is hidden within these scopes. This may lead to ignorance of data management and thereby the unawareness of data quality problem [8]. In Ethiopia, data quality and reliability issues are not well guiding program decisions in all aspects. Poor data quality at the lower administrative level or peripheral levels of woreda and health facilities, which are the source for majority of data used for decision making in the health sector remains a challenge as reported in 2016 annual reports of health sector transformation plan [9].

According to the assessment conducted on HMIS data quality and information use showed content completeness, reporting timeliness and accuracy were 39%, 73% and 76% respectively. Existing evidence shows in Ethiopia including SNNPR (South Nation Nationality People Region) low level of data quality was reported as a gap which was below the national standard. Data accuracy level for health centers was 36.22% which was much lower than the national target. This is due to many factors like lack of training, lack of decision based on supervision, lack of feedback, data quality assurances are done less frequently, limited skills of the health professionals [6, 7, 10, 11].

Even though, as reported on the 2016 annual HSTP performance report of SNNPR, improvements have been seen in HMIS performance in the region, there is still a challenge in data quality especially on indicators related with HIV/AIDS, TB (Tuberculosis) and ANC (Antenatal care). [12]. The annual report of Hadiya Zone in 2017 shows there was a gap in completeness and timeliness of reports. The LQAS (Lot Quality Assurances System) assessment result also show discrepancy of the reports for accuracy of data, over and under reporting of results and does not much expected level of RDQA (Routine Data Quality Assessment)

proportion (0.90–1.10) [13]. Thus, this study aimed to assess the level of data quality and factors associated with data quality in the area.

## Method and materials

### Study setting, study design and study period

This study was conducted in Hadiya Zone which is found in the Southern, Nations, Nationalities and Peoples'Regional State of Ethiopia. Hadaya zone comprises of 10 districts, 2 town administration and 333 kebeles (305 rural kebeles and 28 urban kebeles). Its capital is Hosanna town which is located 205 KM away from Addis Ababa. The Zone is bordered by Gurage Zone in the North, Kembata Tembaro Zone & Halaba special district in the South, Silte Zone in the East and Yem Special district & Omo River in the west. It has one general hospital, 2 primary hospitals, 61 health centers and 309 health posts. At the time of the study there were 2,716 health professionals of different disciplines [14]. Facility based cross sectional study design was employed from March 15, 2018 –April 15, 2018.

**Sample size determination.** *For accuracy dimensions.* Samples of 18 Health centers were selected to assess data quality. Based on the national HMIS information use and data quality manual, seven to nine data elements from each health center is satisfactory to assess data accuracy [15]. Data elements were selected randomly from top priority indicators at national level. Therefore, seven data elements from the 18 selected health centers were verified. 2 month documents were reviewed to see consistence of selected data elements of by random selection of the months September and November. The accuracy of data elements was determined by Accuracy Ratio (recounted data from the source document or registrations over reported data to the next level) for the respective data element. Lower than 0.90 accuracy ratio indicates over-reporting and higher than 1.10 accuracy ratio indicates under-reporting. Seven data elements, Antenatal care fourth visit, institutional deliveries, Pentavalent third doses, PMTCT coverage, Tuberculosis cure rate, confirmed malaria cases, and Contraceptive accepters rate were selected.

*For completeness and timeliness.* Content completeness was assessed by proportion of filled data elements of reporting formats pertaining to selected months. A tolerance level of 90% was used in grading health centers, which meant that each health center expected to complete at least 90% of data elements on report formats. All data elements of two months HMIS reports were reviewed to assess content completeness of reports. Timeliness also assessed by proportion of facilities with number of reports delivered up to deadline come for the selected two months. A tolerance of 90% was used in grading health centers.

**Sample size and sampling procedure.** Sample size was calculated using single population proportion formula based on the following assumption, 75% of peoples capable of performing HIS tasks in Eastern Ethiopia [8], desired degree of precision was 5%, 95% of confidence interval. These results the sample size of 288 and using a contingency of 5% for non-respondents the final sample size will be 302.

WHO recommended for assessment of health facilities by considering the available funds and human resources, selecting 10%-50% facilities to have representative sample. Among the total 61 health centers in the zone 30% of health centers were selected based on the suggestion [16]. A total of 18 health centers were selected by simple random sampling. The calculated sample size for respondents were proportionally allocated to each health center, then health professionals were also selected randomly using lottery method from each selected health center. Health centers that are functional for more than one year were included whereas Health workers who had less than six month experience were excluded.

**Data collection instrument and procedures.** Data collection tools were adapted from the PRISM (Performance of Routine Information System Management) assessment tools version 3.1 and HMIS user's guideline. The tool is prepared to fit with local context and it mainly contains questions to assess accuracy, completeness and timeliness of HMIS data. Self-administered structured questionnaire containing back ground information of the respondents, organizational, behavioural and technical determinants of data quality in health centers was used [15, 17]. The tool was pretested prior to actual data collection period on 5% of the sample size and they were not included in the actual data collection.

The collected data were checked for the completeness and coded before entry and entered to EPI info version 7 then exported to SPSS version 20 for processing and analysis through descriptive statistics. Incomplete, inconsistent and invalid data were refined properly to get maximum quality of data before, during and after data entry. Percentage, Frequency distribution tables and figures were used to describe the study variable for assessment of HMIS.

Binary logistic regression was used to identify the association between problems in data quality and the factors. Bivariable analysis was conducted and variables with $p < 0.25$ selected as candidate variables for multivariate analysis. Finally variables with $p < 0.05$, during multivariable analysis was considered as significant. The overall data quality was calculated by taking the sum of completeness, timeliness and accuracy scores.

The dependent variable were HMIS data quality while the following factors were included in the model as independent variables: Socio-demographic Factors: Age, Sex, Education level, Position of respondents, Work experience: Technical factors;-Complexity of the reporting formats and procedures, Availability of Computer software's (data base), Standard set of indicators with definition.: Individual behavioural factors:- Knowledge of content of HMIS form, Confidence levels for HIS Tasks, Data quality checking skill, Motivation, incentives: Organizational factors;- Management support for HMIS, Training, Supervision, Regular feedback.

## Data quality management

To ensure the quality of data the following activities were done: adapting questionnaires from Standard tools, then translated in to Amharic. Training was given to data collectors on sampling procedures, techniques of interview and data collection process and supervised by the principal investigator. Pre testing of questionnaire was undertaken to check the understandability by taking 5% of sample from other health centers which are not included in the actual data collection. Inconsistent and incomplete data were managed accordingly before data entry in computer software's.

**Variable measurement. Data accuracy;**-was measured by calculating the number from source document over the number from report submitted to the next level. Based on 10% tolerance for data accuracy was classified as follows;- Over reporting ($<0.90$), Acceptable limit ($0.90–1.10$) and Under reporting ($>1.10$).

**Content completeness** was measured by the number of cells of report form which are left blank without indicating "zero". If greater than or equal to 90% of cells of the report filled was considered as complete.

**Report timeliness** was measured by the number of reports delivered up to deadline for facility head over the number of reports expected to come.

**Level of Knowledge:** A health professional said to be knowledgeable if they responds knowledge questions above respondent mean score.

**Confidence level or Self-efficacy;**-was measured in a scale of 0–100 that means from no confidence (zero) to full confidence (100) to perform HMIS tasks.

## Ethics approval and consent to participate

The ethical approval for this study was obtained from the research ethical committee of school of public health, Addis Ababa University; permission letter was written for AA, RHB, Hadiya zone health office, woreda health office and health centers. Then informed written consent was obtained from the participants, after the necessary explanation about the purpose, procedures, benefits, risks of the study is explained and also their right on decision of participating in the study. After getting informed consent from the respondents the right of the respondents to refuse answer for few of all of the questions was respected.

# Result

## Characteristics of respondents

A total of 291 respondents were participated in study with response rate of 96%. Eleven health centers head (3.8%), 137 department heads (47%), 15 HMIS focals (5.2%) and 128 Nurses (44%) were participated in the study. Most of the respondent's age was within the range of 21-30(71.1%). Among the respondents 62.5% were male. Regarding distribution of level of education 190 (65.3%) were level four diploma holders and 101 (34.7%) bachelor degree holders. About 56.7% the respondents were nurses with the maximum experience of 10 years and average experiences of 5 years (Table 1).

## General structure and capability of HMIS

All health centers assigned HMIS focal persons who are responsible for reviewing and aggregating numbers prior to submission to the next level. About 11 health centers assigned HMIS focals who are engaged on other responsibility like service provision. Only 11% of facilities assigned HIT professionals.

**Table 1. Socio demographic characteristics of respondents in health centers of Hadiya zone, Southern Ethiopia, 2018 [n = 291].**

| Variables | Category | Frequency | Percent (%) |
|---|---|---|---|
| Sex of respondents | Male | 182 | 62.5 |
| | Female | 109 | 37.5 |
| Age of respondents | 20–24 | 35 | 12.1 |
| | 25–29 | 139 | 47.9 |
| | 30–34 | 71 | 24.4 |
| | 35–39 | 25 | 8.6 |
| | 40–44 | 21 | 7.2 |
| Educational status | Diploma | 190 | 65.3 |
| | Bachelor degree | 101 | 34.7 |
| Years of experience | 1–5 | 182 | 62.5 |
| | 6–10 | 109 | 37.5 |
| Position of respondents | head of health centers | 11 | 3.8 |
| | department heads | 137 | 47 |
| | HMIS focals | 15 | 5.2 |
| | Nurses | 128 | 44 |
| | Midwife | 45 | 15.5 |
| | Health officer | 46 | 15.8 |
| | Laboratory technician | 20 | 6.9 |
| | Pharmacy | 13 | 4.5 |
| | HIT | 2 | 0.7 |

**Table 2. General structure and capability of HMIS in health centers of Hadiya Zone, southern, Ethiopia 2018.**

| Variables | Expected No- of items | Observed No- of items | % |
|---|---|---|---|
| HMIS focal person | 18 | 18 | 100 |
| have written job descriptions | **18** | 0 | 0 |
| Electronic data base (computer software) | 18 | 5 | 28 |
| currently functional computer software | 18 | 4 | 22 |
| Rules to prevent unauthorized changes to data | 18 | 4 | 22 |
| Establish performance monitoring team | 18 | 18 | 100 |

Based on the finding only 4 health centers were using functional computer software and all have Rules to prevent unauthorized changes to data (password). All 18 health centers were established performance monitoring team (Table 2).

## Record keeping

All health centers kept copies of reports. The count for one year period of copies of reports shows that the monthly report kept ranges from 10–12. From all health centers assessed 96% kept copy of monthly reports that are sent to the next level.

## Accuracy of data

A total of 18 health centers were studied for data quality by accuracy, completeness and timeliness dimensions. Seven data items or indicators were assessed for data accuracy. Service delivery reports and registration books were checked for the month September and November by random selection of the months. Seven indicators verified were Antenatal care fourth visit (ANC 4), Contraceptive acceptance rate (CAR), Institutional delivery, Pentavalent third doses (Penta 3), PMTCT, TB cure rate and confirmed malaria cases from top priority indicators at national level.

From 18 facilities observed 44% of facilities were within acceptable level of accuracy. Data were over reported in all facilities. ANC4 and PMTCT data was over reported by 14 health centers (78%). About 11% health centers under reported TB cure rate and confirmed malaria cases. 14 health centers over reported. Only three out of seven (42.8%) indicators were within 10% acceptable level. About 19% of ANC4 data, over reported (>10% tolerance level) followed by 16%, 15% and 14% CAR, Penta3 and PMTCT data were over reported (>10%). The overall accuracy of data was 76%./

## Completeness of data

Content completeness was assessed by checking two months service delivery report whether the required data elements in a report form are filled or data are complete. Overall content completeness was 83.3%.

## Timeliness of data

Timeliness of the HMIS data was assessed by checking whether HMIS data reporting by the health facilities met the predetermined deadline of reporting period received by the facility head. Over all timeliness was 88.42%. About 55.5% facilities found within 90% tolerance level"-Fig 1".

Based on the three dimensions of data quality which are accuracy, completeness and timeliness the overall data quality of the health centers was 82.5%.

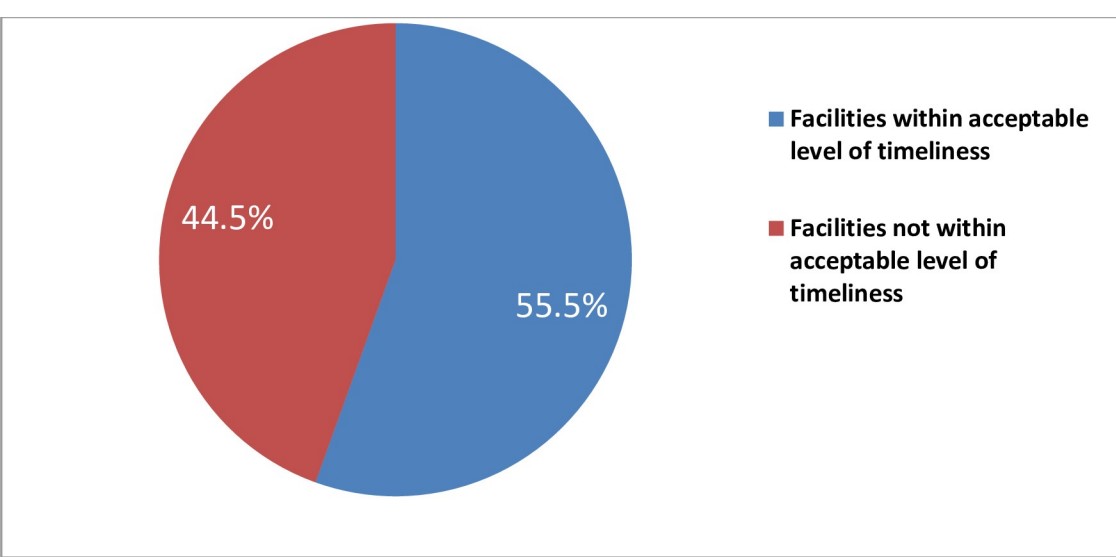

**Fig 1. Timeliness of reports in health centers of Hadiya zone, Southern Ethiopia 2018 Supporting information.**

**HMIS process.** Concerning participation of respondents in HMIS activities among the respondents 87.3% participate in aggregation or compilation of data from registration. More than half the of respondents 57.7% reported that they conduct data quality check but frequency of conducting data quality varied among respondents that about 51.8% conduct data quality test on monthly basis. Overall 86.9% of the respondents reported that they fill registration or tally sheet completely.

## Technical and behavioural factors

From total respondents 59.8% of respondents were reported that they had standard set of indicators including case definitions in their departments. Among the respondents 40.5% reported that there are skilled staff able to aggregate data and to fill out formats and 77.7% reported that HMIS is user friendly format Individual behaviour factors were assessed through individual perception (motivation) towards HMIS use, knowledge of respondents regarding HMIS, confidence level of respondents to do HMIS tasks and availability of incentives for HMIS for HMIS activities. About 28% of respondents reported that availability of incentives for HMIS activity which is training opportunity. About 60.8% of respondents had knowledge towards HMIS. About 66% reported on data quality checking skill and average confidence level of respondents was 63%. Average perception (motivation) of individuals towards HMIS use and meaning was 49.1% (Table 3).

**Self-efficacy.** Confidence level to perform HMIS tasks for health professionals were assessed on a scale of 0 to 100. The average score obtained for the seven questions expressed as a percentage. Higher confidence was observed in checking data accuracy and calculating percentages (66%) and lower confidence was observed in explaining findings from bar charts (56%) relatively. The average confidence level to perform HMIS activities of respondents were 63%.

## Organizational factors

Regarding training status, from the total respondents 52.2% reported that they had received training on HMIS activities. Among those 35.1% took in-service training related with HMIS

**Table 3. Technical and behavioural factors of HMIS data quality in health centers of Hadiya zone, southern Ethiopia 2018.**

| Technical and behavioural factors | Yes (%) | No % |
|---|---|---|
| Standard set of indicators including case definitions | 174(59.8) | 117(40.2) |
| Skilled staff able to aggregate data and to fill out formats | 118 (40.5) | 173(59.5) |
| Complexity of HMIS formats(user friendly format) | 226 (77.7) | 65 (22.3) |
| Incentives | 82 (28) | 209 (72) |
| Knowledge on HMIS | 177(60.8) | 114(39.2) |
| Data quality checking skill | 192(66) | 99 (34) |
| Individual perception(motivation) | 143 (49.1) | 148 (50.9) |
| Self-efficacy (confidence level) | 183 (63) | 108(37) |

tasks. From total respondents 62.5% of respondents supervised one times in last three months from higher officials regarding data quality. Regarding feedback, 55.3% of respondents received feedback from next higher official's among those 60.2% get feedback reports monthly. About 60.8% of respondents agreed on extent of management support regarding HMIS activities.

Among the respondents 61.9% of respondents agreed on, their supervisors give emphasis for data in monthly reports and 55% agreed that supervisors provide regular feedback to their staff. about 63.2% the respondents agreed on, their supervisors check data quality regularly. About 44.3% of respondents agreed on their supervisors encourage over reporting of data for underperformed reports.

**Multivariable analysis.** Variables with p<0.05 taken as predictor of HMIS data quality. Training has shown significant relationship (P<0.05) with data quality [AOR = 1.89, 95% CI (1.03, 3.45)]. Those who were trained 1.89 times more likely to report quality data than who were not trained. Filling registration or formats completely also show significant relationship with data quality [(AOR = 3.4 95% CI (1.3, 8.7)]. Those who fill the registration or formats were 3.4 times more likely report quality data than those who were not fill completely. Self-efficacy (perceived level of confidence) has significant relationship with data quality [AOR = 1.75 95% CI (0.99, 3.11)]. Those who have high level of confidence were 1.75 times more likely to report quality data than those who have low confidence level. Supervision has significant relationship with data quality [AOR = 1.7 95% CI (1.00, 2.95)]. Those supervised health workers were 1.7 times more likely to report quality data compared to who were not supervised. Checking data quality also has significant relationship with data quality [AOR = 1.8 95% CI (0.49, 3.09)]. Those health workers who conduct data quality check were 1.8 times more likely to report quality data compared to who were not (Table 4)

## Discussion

Quality of data is a key factor in generating reliable health information that enables monitoring progress and making decisions for continuous improvement [7]. However the quality of data in the zone based on accuracy, completeness and timeliness showed 76%, 83.3% and 88.4% respectively. Overall data quality of the zone scored 82.5% which was below the national target 85% [5].

All decision of the health system depends on the availability of timely, accurate, and complete information. However the study found 76% of data accuracy. The finding was comparable with the assessment done in Ethiopia, 76% of data accuracy level reported [11]. However According to the baseline assessment done in SNNPR, 36.22% of data accuracy was observed at health centers which was lower than the current study [6]. This may be due to the time gap,

**Table 4. Multi variable logistic regression result on data quality for health centers of Hadiya zone southern Ethiopia 2018.**

| Variables | | data quality | COR (95% CI) | AOR (95% CI) | P- value |
|---|---|---|---|---|---|
| Knowledge on HMIS | Yes | 177(60.8%) | 1.99(1.18,3.35) | 1.209(0.29,2.72) | 0.84 |
| | No | 114(39.2%) | 1 | | |
| filling registration or tally completely | Yes | 253(87%) | 4.42(2.2,8.9) | 3.41*(1.3,8.7) | 0.043 |
| | No | 38(13%) | 1 | | |
| Supervision | Yes | 182(62.5%) | 1.56(0.92,2.63) | 1.71*(1.00,2.95) | 0.037 |
| | No | 109(37.5%) | 1 | | |
| Training | Yes | 152(52.2%) | 1.59(0.95,2.67) | 1.89*(1.03,3.45) | 0.014 |
| | No | 139(47.8%) | | | |
| Confidence level | Confident | 183(63%) | 1.71(1.01,2.9) | 1.75*(0.99,3.11) | 0.047 |
| | Not Confident | 108(37%) | 1 | | |
| Data quality check | Yes | 168(57.7%) | 1.78(1.06,2.9) | 1.8*(0.49,3.09 | 0.032 |
| | No | 123(42.3%) | 1 | | |
| Complexity of the formats | Yes | 226(77.7%) | 1.69(0.94,3.04) | 0.70(0.32,1.50) | 0.36 |
| | No | 65(22.3%) | 1 | | |
| Management support | Yes | 177(60.8%) | 1.99(1.18,3.35) | 0.89(0.29, 2.71) | 0.84 |
| | No | 114(39.2%) | 1 | | |
| Availability of procedural manual | Yes | 146(50.2%) | 1.52(0.908,2.54) | 1.41(0.82,2.44) | 0.22 |
| | No | 145(49.8%) | 1 | | |
| Sense of responsibility | Yes | 175(60.2%) | 2.05(1.22,3.44) | 1.33(0.43,4.13) | 0.62 |
| | No | 116(39.8%) | 1 | | |
| Standard set of indicator | Yes | 174(59.8%) | 1.87(1.69,4.84) | 2.10(0.77,5.73) | 0.144 |
| | No | 117(40.2%) | 1 | | |
| Educational status | Diploma | 190 (65.3%) | 1 | | |
| | Degree | 101(34.7%) | 1.65(0.94,2.90) | 1.52(0.84,2.74) | 0.16 |

* p- value <0.05

COR- Crude odds ratio, AOR- Adjusted odds ratio

7 years between the studies. Out of 18 health centers 8 (44%) health centers were in acceptable level of data tolerance. This finding was supported by the study done in India, 63% facilities were not in acceptable limit of data accuracy [18].

Discrepancy of data was observed in all facilities, what is on register and on report formats. Tendencies of over reporting in all indicators and under reporting of some indicators were observed. The finding was similar with an evaluation done in Tigray region [19]. This may be due to incompleteness of data, not understanding the definition of cases or data elements, or data may not fall within the reporting period [15].

Data were over reported in all facilities. ANC4 and PMTCT data was over reported by 14 health centers (78%). This is supported by a national assessment done by EPHI. From the indicators assessed over reporting was observed in ANC and FP services. The study showed only 30% of ANC data reported was matched with source document but in this study about 88% of ANC4 data was matched. The improvement may be due to the study was nationwide so that including many institutions probably increase inclusion of those facilities with low data quality. Delivery data were over reported about 8% which was similar with EPHI data over reporting >10% [20].

About 11% of health centers under reported TB service data and confirmed malaria cases. PMTCT and ANC data was over reported by 14 health centers. From the indicators assessed, only three out of seven (42.8%) indicators were within 10% acceptable level. About 19% of

ANC4 data, over reported (>10% tolerance level) followed by 16%, 15% and 14% CAR, Penta3 and PMTCT data were over reported (>10%). About 39% of health centers over reported delivery data. This was also comparable with EPHI national assessment where Proportions of public facilities made greater than 10% over (20%) of Penta3 data, 88% PMTCT data was the best-matched data among all indicators [20]. This may be due to the fact that the indicators are from the top priority indicators at national level and needed to be performed well which might lead the facilities to over report and it may also be due to manual entry of data. According to the new information revolution every facility expected to use electronic HMIS but in the studied facilities only four facilities use functional electronic HMIS software (data base).

Regarding content completeness the result found 83.3% of completeness based on 90% tolerance, which was slightly higher than a study conducted in Ayder referral hospital 78.6% and a systematic review conducted in Ethiopia [11, 21]. Whereas the result was comparable with a study conducted previously in the same setting on HMIS utilization 82.8% [22].

Another dimension of data quality was timeliness which is measured by, facilities receiving case teams' reports by the predetermined deadlines. Overall timeliness scored 88.4% based on 90.0% tolerance of timeliness which was higher result from study done in SNNPR 77% [6, 11]. The result also revealed better achievement when compared to study conducted previously in the same setting, only 59.6% reports submitted on recommended time period [13].

Content completeness and timeliness dimensions showed less achievement from a study done in Tigray region and Rwanda where 100% facilities met 90% data tolerance [19, 23]. Possible reasons may be due to lack of knowledge of respondents about the implications of an incomplete data on a report formats and to send reports on timely manner among the health workers and it may also be less emphasis was given for data quality during supervision.

Odds of data quality on those health workers who were filling the source document (registration or tally), higher than those who were not filled [AOR = 3.4, 95% CI (1.3, 8.7)]. Similar finding was found on a studies done in Jimma and Bahir Dar town [24, 25]. This may be due to non understandability (complexity) of the tools/formats, using of untrained workers or shortage of training supports on the forms and registers. So that it is difficult to register all relevant information in correct manner and retrieval of these data will be trouble full.

Concerning supervision, regular Supportive supervision with feedback is a key in addressing quality issues by helping to improve overall performance of HMIS especially for better achievement of data quality [26]. More than half (62.5%), health centers participated in this study supervised by their respective higher level as per standard in the last two quarters. The result was supported by studies conducted previously in Dire Dawa and SNNPR [6, 10]. Even though the result was comparable with other studies conducted earlier, about 37.2% health centers were not supervised regularly. One of the most important mechanisms to improve quality of data is regular supervision. Lack of regular systems on supportive supervision affects the importance and quality of data collection. Therefore without regular and program specific supportive supervision it is difficult to achieve information transformation.

Regarding training, continuous training on HIS activity is important to create awareness and to have trained staff and skilled human resources that are confident and motivated to perform HIS tasks [24]. This study found about 52% of health workers trained regarding HMIS activities. This finding was comparable with other studies done in Dire Dawa 52.7% and South Africa 58% were not trained related with HMIS activities [25, 27]. All health workers who participate in the collection at various sections of healthcare, need continuous capacity building to conduct quality review of RHIS at every stage for in-depth understanding of the stages where quality of data can occur [26, 27]. In this study all focal persons and department heads trained regarding HMIS activities but others, service providers who were not trained were involved in the process of HMIS. This may affect the quality of data.

Odds of health information data quality among Health workers those who were confident enough to perform HMIS activities were higher than those who were not confident [AOR = 1.75, 95% CI (0.99, 3.11)]. The result was supported by studies conducted in SNNPR and South Africa [6, 25]. This factor also suggested by WHO measure evaluation as one determinant of data quality [17]. This may be due to complexity of the formats/tools. If data collection forms are complex to fill in, it affects confidence levels and motivation of data collector [17].

Concerning data quality check, good data management require data quality check at all stages. The checking of data quality is the responsibility of all health workers participating in the data management [28]. In this study about 57.7% of health workers check data quality with a frequency of 51.8% on monthly basis. This is supported by different literatures in done by WHO measure evaluation and a study done in Kenya. From a study done in Kenya about 63% of respondents check data quality but the frequency of carrying out the checks was varying from one respondent to another with majority indicating every quarterly 22% [17, 22, 28].

## Conclusions

Data quality for the three dimensions was 82.5% which is lower than the national target 85% for data accuracy. Over reporting of data was observed at all facilities. About 39% of health centers over reported delivery data. About 9% data of ANC4 over reported (>10% tolerance level) followed by 6%, 5% and 4% CAR, Penta3 and PMTCT data were over reported (>10%). Decisions made using inaccurate, incomplete and reported not on timely manner can affect the health system performance. It was observed that there were inadequacy of supervision, training, HIT professionals, written feedback and procedural manuals. The major factors that affect quality of data were, filling registration or tally completely, training, supervision, data quality check and confidence level. Computerized HMIS data base should be distributed for those who are not using; as it will help to improve data accuracy, timeliness of report and reduce the burden of data collectors.

## Supporting information

**S1 File. Questinnannire English version.**
(DOCX)

**S2 File. Questinnaire Amharic version.**
(DOCX)

## Acknowledgments

Our gratitude goes to supervisors, data collectors, respondents, Hadiya zone health department.

## Author Contributions

**Conceptualization:** Mastewal Solomon, Mesfin Addise, Bahailu Balcha.

**Data curation:** Mastewal Solomon, Mesfin Addise, Bahailu Balcha.

**Formal analysis:** Mastewal Solomon, Bahailu Balcha.

**Funding acquisition:** Mastewal Solomon, Berhan Tassew.

**Investigation:** Mastewal Solomon, Mesfin Addise, Amene Abebe.

**Methodology:** Mastewal Solomon, Mesfin Addise, Berhan Tassew, Bahailu Balcha, Amene Abebe.

**Project administration:** Berhan Tassew.

**Software:** Amene Abebe.

**Supervision:** Mastewal Solomon, Mesfin Addise, Berhan Tassew, Bahailu Balcha, Amene Abebe.

**Validation:** Mastewal Solomon, Mesfin Addise, Amene Abebe.

**Visualization:** Mastewal Solomon, Mesfin Addise.

**Writing – original draft:** Mastewal Solomon, Berhan Tassew, Bahailu Balcha.

**Writing – review & editing:** Mastewal Solomon, Mesfin Addise, Berhan Tassew, Amene Abebe.

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
