## [Decision Letter · Decision Letter 0]

19 Mar 2021

PONE-D-21-05517

Assessment of Quality of data and associated factors in the Health Management Information System among Health Centers of Hadiya Zone, Southern Ethiopia

PLOS ONE

Dear Dr. Bachore,

Thank you for submitting your manuscript to PLOS ONE. After careful consideration, we feel that it has merit but does not fully meet PLOS ONE’s publication criteria as it currently stands. Therefore, we invite you to submit a revised version of the manuscript that addresses the points raised during the review process.

Please submit your revised manuscript. If you will need significantly more time to complete your revisions, please reply to this message or contact the journal office at plosone@plos.org. Please include the following items when submitting your revised manuscript:

We look forward to receiving your revised manuscript.

Kind regards,

Frederick Quinn

Academic Editor

PLOS ONE

Journal Requirements:

We suggest you thoroughly copyedit your manuscript for language usage, spelling, and grammar. If you do not know anyone who can help you do this, you may wish to consider employing a professional scientific editing service. 

Please include in your Methods section (or in Supplementary Information files) the participating hospitals/institutions.

Please include additional information regarding the survey or questionnaire used in the study and ensure that you have provided sufficient details that others could replicate the analyses. For instance, if you developed a questionnaire as part of this study and it is not under a copyright more restrictive than CC-BY, please include a copy, in both the original language and English, as Supporting Information.

5a) If there are ethical or legal restrictions on sharing a de-identified data set, please explain them in detail (e.g., data contain potentially identifying or sensitive patient information) and who has imposed them (e.g., an ethics committee). Please also provide contact information for a data access committee, ethics committee, or other institutional body to which data requests may be sent.

5b) If there are no restrictions, please upload the minimal anonymized data set necessary to replicate your study findings as either Supporting Information files or to a stable, public repository and provide us with the relevant URLs, DOIs, or accession numbers. Please see http://www.bmj.com/content/340/bmj.c181.long for guidelines on how to de-identify and prepare clinical data for publication. For a list of acceptable repositories, please see http://journals.plos.org/plosone/s/data-availability#loc-recommended-repositories.

6. Thank you for stating the following in the Funding Section of your manuscript:

This study has been sponsored by Addis Ababa University

The data will be available on request.

„The funders had no role in study design, data collection and analysis, decision to

publish, or preparation of the manuscript.”

7. Please amend the manuscript submission data (via Edit Submission) to include author Berhan Tassew.

8. We note you have included a table to which you do not refer in the text of your manuscript. Please ensure that you refer to Tables 1-5in your text; if accepted, production will need this reference to link the reader to the Tables.

10. PLOS requires an ORCID iD for the corresponding author in Editorial Manager on papers submitted after December 6th, 2016. Please ensure that you have an ORCID iD and that it is validated in Editorial Manager. To do this, go to ‘Update my Information’ (in the upper left-hand corner of the main menu), and click on the Fetch/Validate link next to the ORCID field. This will take you to the ORCID site and allow you to create a new iD or authenticate a pre-existing iD in Editorial Manager. Please see the following video for instructions on linking an ORCID iD to your Editorial Manager account: https://www.youtube.com/watch?v=_xcclfuvtxQ

11. Thank you for submitting the above manuscript to PLOS ONE. During our internal evaluation of the manuscript, we found significant text overlap between your submission and the following previously published works, some of which you are an author.

http://repository.iifphc.org/handle/123456789/808

Please revise the manuscript to rephrase the duplicated text, cite your sources, and provide details as to how the current manuscript advances on previous work. Please note that further consideration is dependent on the submission of a manuscript that addresses these concerns about the overlap in text with published work.

We will carefully review your manuscript upon resubmission, so please ensure that your revision is thorough

Reviewers' comments:

Reviewer's Responses to Questions

**Comments to the Author**

1. Is the manuscript technically sound, and do the data support the conclusions?

Reviewer #1: Yes

Reviewer #2: Yes

2. Has the statistical analysis been performed appropriately and rigorously? 

Reviewer #1: Yes

Reviewer #2: Yes

3. Have the authors made all data underlying the findings in their manuscript fully available?

Reviewer #1: Yes

Reviewer #2: Yes

4. Is the manuscript presented in an intelligible fashion and written in standard English?

Reviewer #1: Yes

Reviewer #2: Yes

5. Review Comments to the Author

Reviewer #1: For the revision of the document, there are things to complete and others to correct concerning the form and the statistical part.

Reduce the introduction, it is 2 full pages long.

The reference 30 is quoted in the discussion part of the document, but it does not exist in the reference part.

And finally add other recommendations.

Reviewer #2: Please read the instructions given in attached file namely ''Comments and recommandations".

DEAR CORRESPONDING AUTHOR/CO-AUTHORS, PLEASE READ BELOW MESSAGE CAREFULLY.

1. Please, rephrase the title. Instead of “Assessment of quality of data” it’s better to adopt the expression “Data quality assessment and associated factors in the Health Management Information System among Health of Hadiya Zone, Southern Ethiopia".

2. The Abreviations must be writing full, at least once, before putting them as abbreviations ( e.g. SNNPR/EC/HIV/AIDS/TB/ANC/LQAS/RDQA.)

3. Please, erase all extra spaces between words. Ans warning for ponctuations, too. The comma is important, as well as others punctuations (e.g. Data quality was assessed using accuracy, completeness and timeliness)

4. Introduction must be divided into 3 paragraphs : 1. introduction 2. significance of the study and 3. aim of the study.

5. For the results given in the article, the comma in English is not used in maths. (E.g. 23.52 and not 23, 52%)

6. Please insert other references to support your ideas (In introduction for example: (e.g. Therefore the health sector transformation plan (HSTP) considered a need for information revolution as one of the four transformation plan agendas which involves advancement on the methods starting from data collection to the use of information for decision). In this sentence, there are not references.

7. Please, review your English and avoid repeating the expression "In this zone" each time (E.g. …and after the scale up the reformed HMIS was implemented in 2011 in this zone. Also (E.g. However, limited researches are done that can show specifically on the level of HMIS data quality and factors affecting data quality in the region).

8. Methods : Please, correct the way of writing dates.

9. Methods: Authors can opt for a map, instead of a text for geographic localization or rephrase the paragraph.

10. Please, reformulate this sentence: (The study utilized Facility based …..health workers involved in RHIS activities).

11. The steps in methods are to be restructured. You can combinate between some points. In addition, there are many repetitions.

12. This forms (►) are not accepted in scientific articles. Another format will be more professional

13. For the results part, please reclassify the axes, and summarize them so that the information will be clearer.

14. There are empty boxes in your table 5.

15. Tables are also to be combined

16. The form of references is not that requested by PLOSONE revue

6. PLOS authors have the option to publish the peer review history of their article (what does this mean?). If published, this will include your full peer review and any attached files.

Reviewer #1: No

Reviewer #2: No

---

## [Author Response · Author response to Decision Letter 0]

27 May 2021

Response to reviewers

First of all, we would like to acknowledge the academic editor for giving us adequate time to revise and address all the concerns of the reviewers and journal requirements. Following, we the authors of this manuscript have been working extensively since we have been notified with the academic editor and expert reviewer’s report of the manuscript giving a due attention for all the concerns raised by the academic editor and expert reviewers to be well addressed. Thank you so much! 

A. Point by point response letter to academic editor 

1. We have checked again our manuscript for fulfillment of PLOSONEs style requirements including the naming of files and it has been written accordingly. Thank you!!

2. Regarding the language usage, grammar and spelling errors, we have re-written each and every statement throughout the manuscript correcting all the grammatical, spelling and punctuation errors consulting English language professionals. Thank you!

3. According to the academic editor’s recommendation to include the participating health facilities in the methods or as the supplementary file, the participating health institutions are included as supplementary file. Thank you!

4. As per the request of the academic editor we have included all the survey questionnaire (in the original language and the English version) as supporting information. Thank you. Thank you in advance!!!

5. Regarding the data availability all the data used are included in the manuscript. Thank you! 

6. According to the academic editor’s recommendation regarding publicity of the funding source, we have removed the statement from the acknowledgment section in the current version Thank you in advance!!!

7. Regarding inclusion of the author Berhan Tessaw in the online submission, we have included (via edit submission) online and included one additional author (Amene Abebe) who was missed in the first submission. Thank you!

8. The comment regarding referring the tables 1-5 in the text of the manuscript, has been well accepted and we have corrected all the tables in the manuscript text to refer the tables sequentially as they appear in the manuscript text. Thank you!

9. Regarding to check whether retracted articles are still cited in the manuscript, we have checked the references and no retracted reference is used. Thank you!

10. According to the requirement for the ORCID ID, the corresponding author and some of co-authors such as (Amene Abebe) already had ORCID iD. Thank you!

11. The comment about overlapping texts with previous publication is correct. However, nothing was deliberate because the publications which I were a co-author were published in the year 2020 and the current study has been done in the year 2018. Though, we have rephrased the overlapping texts in the current version of the manuscript. Thank you! 

B. Point by point response letter to reviewer one

1. Concerning the reviewer comment about revision of the document, form and the statistical part. We have re-written the document and corrected all errors in the write up and statistical output reporting 

2. The reviewer comment about reduction of the introduction section of the manuscript is well accepted. We have reduced the introduction section to only one page and a paragraph, and restructured the content in to what has been known in the existing literature, what is lacking and what has been aimed by the study. Thank you!

C. Point by point response letter to reviewer two

First of all we would like to express our gratefulness for the reviewer for appropriately recognizing the topic as one of the important area of research and for being interested on the topic. Following our acknowledgement, the reviewers concerns are point by point addressed in the following ways:

1. Based on the recommendation of the reviewer to rephrase the title, we have rephrased the title as per the suggestion. Thank you! 

2. The comment about writing in full the abbreviations at least once before using the abbreviation is well accepted. We have written all the abbreviation in full at least once before we use the abbreviations in the manuscript file. Thank you! 

3. The reviewer comment about erasing unnecessary space between words and punctuation is right. We have erased all the unnecessary spaces and also checked punctuation errors throughout the document and corrected. Thank you!

4. The reviewer comments regarding the structure of the introduction section of the previous version of the manuscript has been accepted. In the current version we re-structured the contents of the introduction in to three sections “introduction” “significance of the study” and “aim of the study” and we have removed all the repeated concepts which unnecessarily lengthened the introduction section of the manuscript. Thank you so much! 

5. The reviewer comment about the use of comma in English is not used in maths is right. However, we search throughout the manuscript document and couldn’t get such error. Thank you!

6. The reviewer comment about insertion of references to support ideas is accepted. We have cited the references as needed throughout the manuscript document. Thank you!

7. The reviewer comment regarding the repetition of words like “the zone” and “the region” unnecessarily have been removed and corrected. Thank you!

8. According to the reviewers comment to correct the way of writing of the dates in the methods section is accepted. We have corrected the writing in the current version. Thank you!

9. The reviewer comment regarding rephrasing or inclusion of the map of the study area is right. We have rephrased and restructured the description of the study setting, study design and study period and made more clear and precise. Thank you! 

10. The sentence in the methods section which written (The study utilized Facility based …..health workers involved in RHIS activities) has been reformulated. Thank you!

11. The reviewer recommendation to restructure and combine the repeated points in the methods section is well accepted. In the recent version we have removed the repeated points and also combined whose idea is the same. Thank you!

12. The comment of the reviewer about not use the bullet (►) is accepted. We have corrected in the current version. Thank you! 

13. The comment about reclassifying of the axes and summarizing the results is accepted. We have summarized and present the results section in a clearer way in this version. Thank you!

14. The reviewers comment about empty boxes in the table 5 is corrected. Thank you!

15. The reviewers comment about small tables is correct and we have combined table 3 and 4.Thank you!

16. The reviewer comment regarding the form of references is not that requested by PLOSONE revue is right and we have corrected in the current version as per the journal reference citation requirement. Thank you!

---

## [Decision Letter · Decision Letter 1]

4 Jun 2021

PONE-D-21-05517R1

Data quality assessment and associated factors in the Health Management Information System among Health Centers of Hadiya Zone, Southern Ethiopia

PLOS ONE

Dear Dr. Balcha,

Thank you for submitting your manuscript to PLOS ONE. After careful consideration, we feel that it has merit but does not fully meet PLOS ONE’s publication criteria as it currently stands. Therefore, we invite you to submit a revised version of the manuscript that addresses the points raised during the review process.

Please submit your revised manuscript. If you will need significantly more time to complete your revisions, please reply to this message or contact the journal office at plosone@plos.org. Please include the following items when submitting your revised manuscript:

We look forward to receiving your revised manuscript.

Kind regards,

Frederick Quinn

Academic Editor

PLOS ONE

Journal Requirements:

Reviewers' comments:

Reviewer's Responses to Questions

**Comments to the Author**

1. If the authors have adequately addressed your comments raised in a previous round of review and you feel that this manuscript is now acceptable for publication, you may indicate that here to bypass the “Comments to the Author” section, enter your conflict of interest statement in the “Confidential to Editor” section, and submit your "Accept" recommendation.

Reviewer #1: All comments have been addressed

2. Is the manuscript technically sound, and do the data support the conclusions?

Reviewer #1: Yes

3. Has the statistical analysis been performed appropriately and rigorously? 

Reviewer #1: Yes

4. Have the authors made all data underlying the findings in their manuscript fully available?

Reviewer #1: Yes

5. Is the manuscript presented in an intelligible fashion and written in standard English?

Reviewer #1: Yes

6. Review Comments to the Author

Reviewer #1: good evening

I thank the authors for their understanding

regarding this version of the manuscript.

I have some recommendations.

1-in the title: I propose the following title: "Data quality assessment and associated factors in the Health Management Information System among Health Centers of Southern Ethiopia".

2-in the part: Study setting, study design and study period

add a geographical map of the study area.

3-you tend to write decimals with "." and "%" at the same time, (0.9 or 90% , 1.10 or 110% ) page 13 and page 15.

4-page 18 paragraph HMIS process ,3 line the number in parenthesis 57.7%, delete the parenthesis.

5-page 23: add "=" to paragraph 4 [AOR = 3.4, 95% CI (1.3, 8.7)].

6-reference 8 is identical to reference 5.

7-the authors of the last two references 28 and 29 are the same authors, so correct the names of the authors, use the same style of citation of references.

7. PLOS authors have the option to publish the peer review history of their article (what does this mean?). If published, this will include your full peer review and any attached files.

Reviewer #1: No

---

## [Author Response · Author response to Decision Letter 1]

17 Jun 2021

Response to academic editors and the reviewer

First of all, we would like to acknowledge the academic editor for giving us adequate time to revise and address all the concerns of the reviewer and journal requirements. We have addressed the recommendations of the reviewer on the second version of the manuscript as follows. Thank you!

Response to the academic editors

• Regarding the journal requirement about the citation of the retracted articles, we have checked the references we used and no retracted article has been cited in in our manuscript. Thank you!

Response to the reviewer

1. As per the recommendation of the reviewer to modify the title as “Data quality assessment and associated factors in the Health Management Information System among Health Centers of Southern Ethiopia” is accepted and we have modified it as recommended. Thank you!

2. The reviewer recommendation to add the geographic map of the study area is accepted and we have added the study area map in the “study setting, study design and study period” section of the current revised manuscript. Thank you!

3. According to the reviewer suggestion not to use decimal ”.”and”%” at the same time with in (0.9 or 90% , 1.10 or 110% ) page 13 and page 15. We have corrected the numbers to be reported only with the decimal numbered. Thank you! 

4. According to the reviewer recommendation to remove the parenthesis from the percentage page 18, 57.7%. We have removed the parenthesis. Thank you!

5. According to the reviewer suggestion to add”=” in page 23 paragraph 4 [AOR = 3.4, 95% CI (1.3, 8.7)]. We have added the equality sign. Thank you!

6. We would like to appreciate the reviewer for showing repetition of reference 5 and 8 and we have removed reference number 8 and corrected the whole document references sequentially. Thank you!

7. Regarding the reviewer comment to maintain consistency of the same authors naming in reference number 28 and 29, the comment is well accepted and we have made the same style of citation in both references. Thank you!

---

## [Decision Letter · Decision Letter 2]

8 Jul 2021

PONE-D-21-05517R2

Data quality assessment and associated factors in the Health Management Information System among Health Centers of Southern Ethiopia

PLOS ONE

Dear Dr. Balcha,

Thank you for submitting your manuscript to PLOS ONE. After careful consideration, we feel that it has merit but does not fully meet PLOS ONE’s publication criteria as it currently stands. Therefore, we invite you to submit a revised version of the manuscript that addresses the points raised during the review process.

Please submit your revised manuscript. If you will need significantly more time to complete your revisions, please reply to this message or contact the journal office at plosone@plos.org. Please include the following items when submitting your revised manuscript:

We look forward to receiving your revised manuscript.

Kind regards,

Frederick Quinn

Academic Editor

PLOS ONE

Journal Requirements:

Additional Editor Comments (if provided):

Reviewers' comments:

Reviewer's Responses to Questions

**Comments to the Author**

1. If the authors have adequately addressed your comments raised in a previous round of review and you feel that this manuscript is now acceptable for publication, you may indicate that here to bypass the “Comments to the Author” section, enter your conflict of interest statement in the “Confidential to Editor” section, and submit your "Accept" recommendation.

Reviewer #1: All comments have been addressed

2. Is the manuscript technically sound, and do the data support the conclusions?

Reviewer #1: Yes

3. Has the statistical analysis been performed appropriately and rigorously? 

Reviewer #1: Yes

4. Have the authors made all data underlying the findings in their manuscript fully available?

Reviewer #1: Yes

5. Is the manuscript presented in an intelligible fashion and written in standard English?

Reviewer #1: Yes

6. Review Comments to the Author

Reviewer #1: At the level of references review correct the following reference:

replace FMoH by FMOH.

26.FMoH. Health Management Information System (HMIS) / Monitoring and Evaluation

(M&E). 2008;

7. PLOS authors have the option to publish the peer review history of their article (what does this mean?). If published, this will include your full peer review and any attached files.

Reviewer #1: **Yes: **ZENIA Safia

---

## [Decision Letter · Decision Letter 3]

28 Jul 2021

Data quality assessment and associated factors in the Health Management Information System among Health Centers of Southern Ethiopia

PONE-D-21-05517R3

Dear Dr. Balcha,

We’re pleased to inform you that your manuscript has been judged scientifically suitable for publication and will be formally accepted for publication once it meets all outstanding technical requirements.

Kind regards,

Frederick Quinn

Academic Editor

PLOS ONE

Additional Editor Comments (optional):

Reviewers' comments:

Reviewer's Responses to Questions

**Comments to the Author**

1. If the authors have adequately addressed your comments raised in a previous round of review and you feel that this manuscript is now acceptable for publication, you may indicate that here to bypass the “Comments to the Author” section, enter your conflict of interest statement in the “Confidential to Editor” section, and submit your "Accept" recommendation.

Reviewer #1: All comments have been addressed

2. Is the manuscript technically sound, and do the data support the conclusions?

Reviewer #1: Yes

3. Has the statistical analysis been performed appropriately and rigorously? 

Reviewer #1: Yes

4. Have the authors made all data underlying the findings in their manuscript fully available?

Reviewer #1: Yes

5. Is the manuscript presented in an intelligible fashion and written in standard English?

Reviewer #1: Yes

6. Review Comments to the Author

Reviewer #1: I thank you for the efforts you have made to respond to our comments.

a work that deserves to be published

Sincerely yours

7. PLOS authors have the option to publish the peer review history of their article (what does this mean?). If published, this will include your full peer review and any attached files.

Reviewer #1: No

---

## [Editor Report · Acceptance letter]

19 Oct 2021

PONE-D-21-05517R3 

Data quality assessment and associated factors in the Health Management Information System among Health Centers of Southern Ethiopia 

Dear Dr. Balcha:

I'm pleased to inform you that your manuscript has been deemed suitable for publication in PLOS ONE. Congratulations! Your manuscript is now with our production department. 

Kind regards, 

on behalf of

Dr. Frederick Quinn 

Academic Editor

PLOS ONE